# Nonlinear random matrix theory for deep learning

**Jeffrey Pennington**
Google Brain
jpennin@google.com

**Pratik Worah**
Google Research
pworah@google.com

## Abstract

Neural network configurations with random weights play an important role in the analysis of deep learning. They define the initial loss landscape and are closely related to kernel and random feature methods. Despite the fact that these networks are built out of random matrices, the vast and powerful machinery of random matrix theory has so far found limited success in studying them. A main obstacle in this direction is that neural networks are nonlinear, which prevents the straightforward utilization of many of the existing mathematical results. In this work, we open the door for direct applications of random matrix theory to deep learning by demonstrating that the pointwise nonlinearities typically applied in neural networks can be incorporated into a standard method of proof in random matrix theory known as the moments method. The test case for our study is the Gram matrix $Y^T Y$, $Y = f(WX)$, where $W$ is a random weight matrix, $X$ is a random data matrix, and $f$ is a pointwise nonlinear activation function. We derive an explicit representation for the trace of the resolvent of this matrix, which defines its limiting spectral distribution. We apply these results to the computation of the asymptotic performance of single-layer random feature networks on a memorization task and to the analysis of the eigenvalues of the data covariance matrix as it propagates through a neural network. As a byproduct of our analysis, we identify an intriguing new class of activation functions with favorable properties.

## 1 Introduction

The list of successful applications of deep learning is growing at a staggering rate. Image recognition (Krizhevsky et al., 2012), audio synthesis (Oord et al., 2016), translation (Wu et al., 2016), and speech recognition (Hinton et al., 2012) are just a few of the recent achievements. Our theoretical understanding of deep learning, on the other hand, has progressed at a more modest pace. A central difficulty in extending our understanding stems from the complexity of neural network loss surfaces, which are highly non-convex functions, often of millions or even billions (Shazeer et al., 2017) of parameters.

In the physical sciences, progress in understanding large complex systems has often come by approximating their constituents with random variables; for example, statistical physics and thermodynamics are based in this paradigm. Since modern neural networks are undeniably large complex systems, it is natural to consider what insights can be gained by approximating their parameters with random variables. Moreover, such random configurations play at least two privileged roles in neural networks: they define the initial loss surface for optimization, and they are closely related to random feature and kernel methods. Therefore it is not surprising that random neural networks have attracted significant attention in the literature over the years.

Another useful technique for simplifying the study of large complex systems is to approximate their size as infinite. For neural networks, the concept of size has at least two axes: the number

of samples and the number of parameters. It is common, particularly in the statistics literature, to consider the mean performance of a finite-capacity model against a given data distribution. From this perspective, the number of samples, $m$, is taken to be infinite relative to the number of parameters, $n$, i.e. $n/m \to 0$. An alternative perspective is frequently employed in the study of kernel or random feature methods. In this case, the number of parameters is taken to be infinite relative to the number of samples, i.e. $n/m \to \infty$. In practice, however, most successful modern deep learning architectures tend to have both a large number of samples *and* a large number of parameters, often of roughly the same order of magnitude. (One simple explanation for this scaling may just be that the other extremes tend to produce over- or under-fitting). Motivated by this observation, in this work we explore the infinite size limit in which both the number of samples and the number of parameters go to infinity at the same rate, i.e. $n, m \to \infty$ with $n/m = \phi$, for some finite constant $\phi$. This perspective puts us squarely in the regime of random matrix theory.

An abundance of matrices are of practical and theoretical interest in the context of random neural networks. For example, the output of the network, its Jacobian, and the Hessian of the loss function with respect to the weights are all interesting objects of study. In this work we focus on the computation of the eigenvalues of the matrix $M \equiv \frac{1}{m} Y^T Y$, where $Y = f(WX)$, $W$ is a Gaussian random weight matrix, $X$ is a Gaussian random data matrix, and $f$ is a pointwise activation function. In many ways, $Y$ is a basic primitive whose understanding is necessary for attacking more complicated cases; for example, $Y$ appears in the expressions for all three of the matrices mentioned above. But studying $Y$ is also quite interesting in its own right, with several interesting applications to machine learning that we will explore in Section 4.

## 1.1 Our contribution

The nonlinearity of the activation function prevents us from leveraging many of the existing mathematical results from random matrix theory. Nevertheless, most of the basic tools for computing spectral densities of random matrices still apply in this setting. In this work, we show how to overcome some of the technical hurdles that have prevented explicit computations of this type in the past. In particular, we employ the so-called *moments method*, deducing the spectral density of $M$ from the traces $\operatorname{tr} M^k$. Evaluating the traces involves computing certain multi-dimensional integrals, which we show how to evaluate, and enumerating a certain class of graphs, for which we derive a generating function. The result of our calculation is a quartic equation which is satisfied by the trace of the resolvent of $M$, $G(z) = -\mathbb{E}[\operatorname{tr}(M - zI)^{-1}]$. It depends on two parameters that together capture the only relevant properties of the nonlinearity $f$: $\eta$, the Gaussian mean of $f^2$, and $\zeta$, the square of the Gaussian mean of $f'$. Overall, the techniques presented here pave the way for studying other types of nonlinear random matrices relevant for the theoretical understanding of neural networks.

## 1.2 Applications of our results

We show that the training loss of a ridge-regularized single-layer random-feature least-squares memorization problem with regularization parameter $\gamma$ is related to $-\gamma^2 G'(-\gamma)$. We observe increased memorization capacity for certain types of nonlinearities relative to others. In particular, for a fixed value of $\gamma$, the training loss is lower if $\eta/\zeta$ is large, a condition satisfied by a large class of activation functions, for example when $f$ is close to an even function. We believe this observation could have an important practical impact in designing next-generation activation functions.

We also examine the eigenvalue density of $M$ and observe that if $\zeta = 0$ the distribution collapses to the Marchenko-Pastur distribution (Marčenko & Pastur, 1967), which describes the eigenvalues of the Wishart matrix $X^T X$. We therefore make the surprising observation that there exist functions $f$ such that $f(WX)$ has the same singular value distribution as $X$. Said another way, the eigenvalues of the data covariance matrix are unchanged in distribution after passing through a single nonlinear layer of the network. We conjecture that this property is actually satisfied through arbitrary layers of the network, and find supporting numerical evidence. This conjecture may be regarded as a claim about the *universality* of our results with respect to the distribution of $X$. Note that preserving the first moment of this distribution is also an effect achieved through batch normalization (Ioffe & Szegedy, 2015), although higher moments are not necessarily preserved. We therefore offer the hypothesis that choosing activation functions with $\zeta = 0$ might lead to improved training performance, in the same way that batch normalization does, at least early in training.

## 1.3 Related work

The study of random neural networks has a relatively long history, with much of the initial work focusing on approaches from statistical physics and the theory of spin glasses. For example, Amit et al. (1985) analyze the long-time behavior of certain dynamical models of neural networks in terms of an Ising spin-glass Hamiltonian, and Gardner & Derrida (1988) examine the storage capacity of neural networks by studying the density of metastable states of a similar spin-glass system. More recently, Choromanska et al. (2015) studied the critical points of random loss surfaces, also by examining an associated spin-glass Hamiltonian, and Schoenholz et al. (2017) developed an exact correspondence between random neural networks and statistical field theory.

In a somewhat tangential direction, random neural networks have also been investigated through their relationship to kernel methods. The correspondence between infinite-dimensional neural networks and Gaussian processes was first noted by Neal (1994a,b). In the finite-dimensional setting, the approximate correspondence to kernel methods led to the development random feature methods that can accelerate the training of kernel machines (Rahimi & Recht, 2007). More recently, a duality between random neural networks with general architectures and compositional kernels was explored by Daniely et al. (2016).

In the last several years, random neural networks have been studied from many other perspectives. Saxe et al. (2014) examined the effect of random initialization on the dynamics of learning in deep linear networks. Schoenholz et al. (2016) studied how information propagates through random networks, and how that affects learning. Poole et al. (2016) and Raghu et al. (2016) investigated various measures of expressivity in the context of deep random neural networks.

Despite this extensive literature related to random neural networks, there has been relatively little research devoted to studying random matrices with nonlinear dependencies. The main focus in this direction has been kernel random matrices and robust statistics models (El Karoui et al., 2010; Cheng & Singer, 2013). In a closely-related contemporaneous work, Louart et al. (2017) examined the resolvent of Gram matrix $YY^T$ in the case where $X$ is deterministic.

## 2 Preliminaries

Throughout this work we will be relying on a number of basic concepts from random matrix theory. Here we provide a lightning overview of the essentials, but refer the reader to the more pedagogical literature for background (Tao, 2012).

### 2.1 Notation

Let $X \in \mathbb{R}^{n_0 \times m}$ be a random data matrix with i.i.d. elements $X_{i\mu} \sim \mathcal{N}(0, \sigma_x^2)$ and $W \in \mathbb{R}^{n_1 \times n_0}$ be a random weight matrix with i.i.d. elements $W_{ij} \sim \mathcal{N}(0, \sigma_w^2/n_0)$. As discussed in Section 1, we are interested in the regime in which both the row and column dimensions of these matrices are large and approach infinity at the same rate. In particular, we define

$$\phi \equiv \frac{n_0}{m}, \quad \psi \equiv \frac{n_0}{n_1}, \tag{1}$$

to be fixed constants as $n_0, n_1, m \to \infty$. In what follows, we will frequently consider the limit that $n_0 \to \infty$ with the understanding that $n_1 \to \infty$ and $m \to \infty$, so that eqn. (1) is satisfied.

We denote the matrix of pre-activations by $Z = WX$. Let $f : \mathcal{R} \to \mathcal{R}$ be a function with zero mean and finite moments,

$$\int \frac{dz}{\sqrt{2\pi}} e^{-\frac{z^2}{2}} f(\sigma_w \sigma_x z) = 0, \qquad \left| \int \frac{dz}{\sqrt{2\pi}} e^{-\frac{z^2}{2}} f(\sigma_w \sigma_x z)^k \right| < \infty \text{ for } k > 1, \tag{2}$$

and denote the matrix of post-activations $Y = f(Z)$, where $f$ is applied pointwise. We will be interested in the Gram matrix,

$$M = \frac{1}{m} YY^T \in \mathbb{R}^{n_1 \times n_1}. \tag{3}$$

## 2.2 Spectral density and the Stieltjes transform

The *empirical spectral density* of $M$ is defined as,

$$\rho_M(t) = \frac{1}{n_1} \sum_{j=1}^{n_1} \delta\left(t - \lambda_j(M)\right), \tag{4}$$

where $\delta$ is the Dirac delta function, and the $\lambda_j(M)$, $j = 1, \ldots, n_1$, denote the $n_1$ eigenvalues of $M$, including multiplicity. The *limiting spectral density* is defined as the limit of eqn. (4) as $n_1 \to \infty$, if it exists.

For $z \in \mathbb{C} \setminus \operatorname{supp}(\rho_M)$ the *Stieltjes transform* $G$ of $\rho_M$ is defined as,

$$G(z) = \int \frac{\rho_M(t)}{z - t} dt = -\frac{1}{n_1} \mathbb{E}\left[\operatorname{tr}(M - zI_{n_1})^{-1}\right], \tag{5}$$

where the expectation is with respect to the random variables $W$ and $X$. The quantity $(M - zI_{n_1})^{-1}$ is the *resolvent* of $M$. The spectral density can be recovered from the Stieltjes transform using the inversion formula,

$$\rho_M(\lambda) = -\frac{1}{\pi} \lim_{\epsilon \to 0^+} \operatorname{Im} G(\lambda + i\epsilon). \tag{6}$$

## 2.3 Moment method

One of the main tools for computing the limiting spectral distributions of random matrices is the moment method, which, as the name suggests, is based on computations of the moments of $\rho_M$. The asymptotic expansion of eqn. (5) for large $z$ gives the Laurent series,

$$G(z) = \sum_{k=0}^{\infty} \frac{m_k}{z^{k+1}}, \tag{7}$$

where $m_k$ is the $k$th moment of the distribution $\rho_M$,

$$m_k = \int dt\, \rho_M(t) t^k = \frac{1}{n_1} \mathbb{E}\left[\operatorname{tr} M^k\right]. \tag{8}$$

If one can compute $m_k$, then the density $\rho_M$ can be obtained via eqns. (7) and (6). The idea behind the moment method is to compute $m_k$ by expanding out powers of $M$ inside the trace as,

$$\frac{1}{n_1} \mathbb{E}\left[\operatorname{tr} M^k\right] = \frac{1}{n_1} \mathbb{E}\left[\sum_{i_1, \ldots, i_k \in [n_1]} M_{i_1 i_2} M_{i_2 i_3} \cdots M_{i_{k-1} i_k} M_{i_k i_1}\right], \tag{9}$$

and evaluating the leading contributions to the sum as the matrix dimensions go to infinity, i.e. as $n_0 \to \infty$. Determining the leading contributions involves a complicated combinatorial analysis, combined with the evaluation of certain nontrivial high-dimensional integrals. In the next section and the supplementary material, we provide an outline for how to tackle these technical components of the computation.

# 3 The Stieltjes transform of $M$

## 3.1 Main result

The following theorem characterizes $G$ as the solution to a quartic polynomial equation.

**Theorem 1.** *For $M$, $\phi$, $\psi$, $\sigma_w$, and $\sigma_x$ defined as in Section 2.1, and constants $\eta$ and $\zeta$ defined as,*

$$\eta = \int dz\, \frac{e^{-z^2/2}}{\sqrt{2\pi}} f(\sigma_w \sigma_x z)^2 \quad and \quad \zeta = \left[\sigma_w \sigma_x \int dz\, \frac{e^{-z^2/2}}{\sqrt{2\pi}} f'(\sigma_w \sigma_x z)\right]^2, \tag{10}$$

*the Stieltjes transform of the spectral density of $M$ satisfies,*

$$G(z) = \frac{\psi}{z} P\left(\frac{1}{z\psi}\right) + \frac{1-\psi}{z}, \tag{11}$$

*where,*

$$P = 1 + (\eta - \zeta)t P_\phi P_\psi + \frac{P_\phi P_\psi t \zeta}{1 - P_\phi P_\psi t \zeta}, \tag{12}$$

*and*

$$P_\phi = 1 + (P-1)\phi, \quad P_\psi = 1 + (P-1)\psi. \tag{13}$$

The proof of Theorem 1 is relatively long and complicated, so it's deferred to the supplementary material. The main idea underlying the proof is to translate the calculation of the moments in eqn. (7) into two subproblems, one of enumerating certain connected outer-planar graphs, and another of evaluating integrals that correspond to cycles in those graphs. The complexity resides both in characterizing which outer-planar graphs contribute at leading order to the moments, and also in computing those moments explicitly. A generating function encapsulating these results ($P$ from Theorem 1) is shown to satisfy a relatively simple recurrence relation. Satisfying this recurrence relation requires that $P$ solve eqn. (12). Finally, some bookkeeping relates $G$ to $P$.

### 3.2 Limiting cases

#### 3.2.1 $\eta = \zeta$

In Section 3 of the supplementary material, we use a Hermite polynomial expansion of $f$ to show that $\eta = \zeta$ if and only if $f$ is a linear function. In this case, $M = ZZ^T$, where $Z = WX$ is a product of Gaussian random matrices. Therefore we expect $G$ to reduce to the Stieltjes transform of a so-called product Wishart matrix. In (Dupic & Castillo, 2014), a cubic equation defining the Stieltjes transform of such matrices is derived. Although eqn. (11) is generally quartic, the coefficient of the quartic term vanishes when $\eta = \zeta$ (see Section 4 of the supplementary material). The resulting cubic polynomial is in agreement with the results in (Dupic & Castillo, 2014).

#### 3.2.2 $\zeta = 0$

Another interesting limit is when $\zeta = 0$, which significantly simplifies the expression in eqn. (12). Without loss of generality, we can take $\eta = 1$ (the general case can be recovered by rescaling $z$). The resulting equation is,

$$z G^2 + \left(\left(1 - \frac{\psi}{\phi}\right)z - 1\right)G + \frac{\psi}{\phi} = 0, \tag{14}$$

which is precisely the equation satisfied by the Stieltjes transform of the Marchenko-Pastur distribution with shape parameter $\phi/\psi$. Notice that when $\psi = 1$, the latter is the limiting spectral distribution of $XX^T$, which implies that $YY^T$ and $XX^T$ have the same limiting spectral distribution. Therefore we have identified a novel type of isospectral nonlinear transformation. We investigate this observation in Section 4.1.

## 4 Applications

### 4.1 Data covariance

Consider a deep feedforward neural network with $l$th-layer post-activation matrix given by,

$$Y^l = f(W^l Y^{l-1}), \quad Y^0 = X. \tag{15}$$

The matrix $Y^l(Y^l)^T$ is the $l$th-layer data covariance matrix. The distribution of its eigenvalues (or the singular values of $Y^l$) determine the extent to which the input signals become distorted or stretched as they propagate through the network. Highly skewed distributions indicate strong anisotropy in the embedded feature space, which is a form of poor conditioning that is likely to derail or impede learning. A variety of techniques have been developed to alleviate this problem, the most popular of which is batch normalization. In batch normalization, the variance of individual activations across the batch (or dataset) is rescaled to equal one. The covariance is often ignored — variants that attempt to

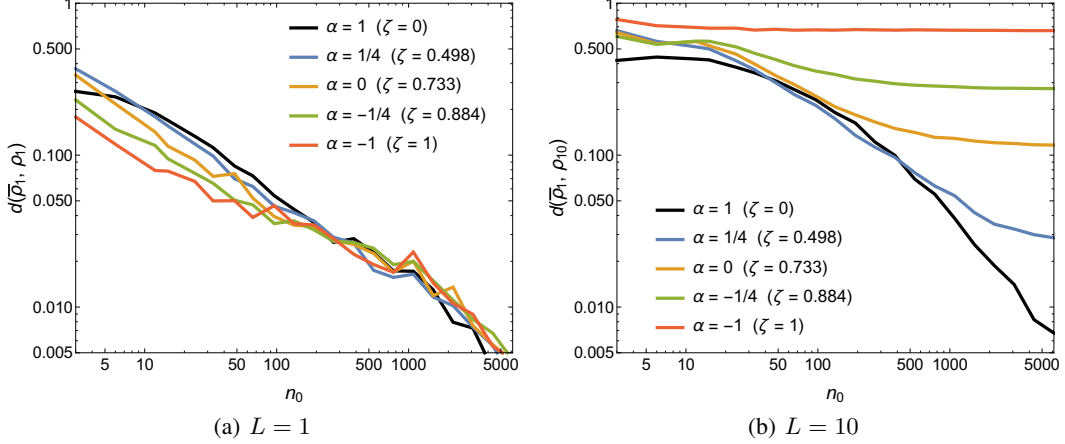

(a) $L = 1$          (b) $L = 10$

Figure 1: Distance between the (a) first-layer and (b) tenth-layer empirical eigenvalue distributions of the data covariance matrices and our theoretical prediction for the first-layer limiting distribution $\bar{\rho}_1$, as a function of network width $n_0$. Plots are for shape parameters $\phi = 1$ and $\psi = 3/2$. The different curves correspond to different piecewise linear activation functions parameterize by $\alpha$: $\alpha = -1$ is linear, $\alpha = 0$ is (shifted) relu, and $\alpha = 1$ is (shifted) absolute value. In (a), for all $\alpha$, we see good convergence of the empirical distribution $\rho_1$ to our asymptotic prediction $\bar{\rho}_1$. In (b), in accordance with our conjecture, we find good agreement between $\bar{\rho}_1$ and the tenth-layer empirical distribution $\zeta = 0$, but not for other values of $\zeta$. This provides evidence that when $\zeta = 0$ the eigenvalue distribution is preserved by the nonlinear transformations.

fully whiten the activations can be very slow. So one aspect of batch normalization, as it is used in practice, is that it preserves the trace of the covariance matrix (i.e. the first moment of its eigenvalue distribution) as the signal propagates through the network, but it does not control higher moments of the distribution. A consequence is that there may still be a large imbalance in singular values.

An interesting question, therefore, is whether there exist efficient techniques that could preserve or approximately preserve the full singular value spectrum of the activations as they propagate through the network. Inspired by the results of Section 3.2.2, we hypothesize that choosing an activation function with $\zeta = 0$ may be one way to approximately achieve this behavior, at least early in training. From a mathematical perspective, this hypothesis is similar to asking whether our results in eqn. (11) are *universal* with respect to the distribution of $X$. We investigate this question empirically.

Let $\rho_l$ be the empirical eigenvalue density of $Y^l(Y^l)^T$, and let $\bar{\rho}_1$ be the limiting density determined by eqn. (11) (with $\psi = 1$). We would like to measure the distance between $\bar{\rho}_1$ and $\rho_l$ in order to see whether the eigenvalues propagate without getting distorted. There are many options that would suffice, but we choose to track the following metric,

$$d(\bar{\rho}_1, \rho_l) \equiv \int d\lambda \, |\bar{\rho}_1(\lambda) - \rho_l(\lambda)| \, . \tag{16}$$

To observe the effect of varying $\zeta$, we utilize a variant of the relu activation function with non-zero slope for negative inputs,

$$f_\alpha(x) = \frac{[x]_+ + \alpha[-x]_+ - \frac{1+\alpha}{\sqrt{2\pi}}}{\sqrt{\frac{1}{2}(1 + \alpha^2) - \frac{1}{2\pi}(1+\alpha)^2}} \, . \tag{17}$$

One may interpret $\alpha$ as (the negative of) the ratio of the slope for negative $x$ to the slope for positive $x$. It is straightforward to check that $f_\alpha$ has zero Gaussian mean and that,

$$\eta = 1, \quad \zeta = \frac{(1-\alpha)^2}{2(1+\alpha^2) - \frac{2}{\pi}(1+\alpha)^2} \, , \tag{18}$$

so we can adjust $\zeta$ (without affecting $\eta$) by changing $\alpha$. Fig. 1(a) shows that for any value of $\alpha$ (and thus $\zeta$) the distance between $\bar{\rho}_1$ and $\rho_1$ approaches zero as the network width increases. This offers

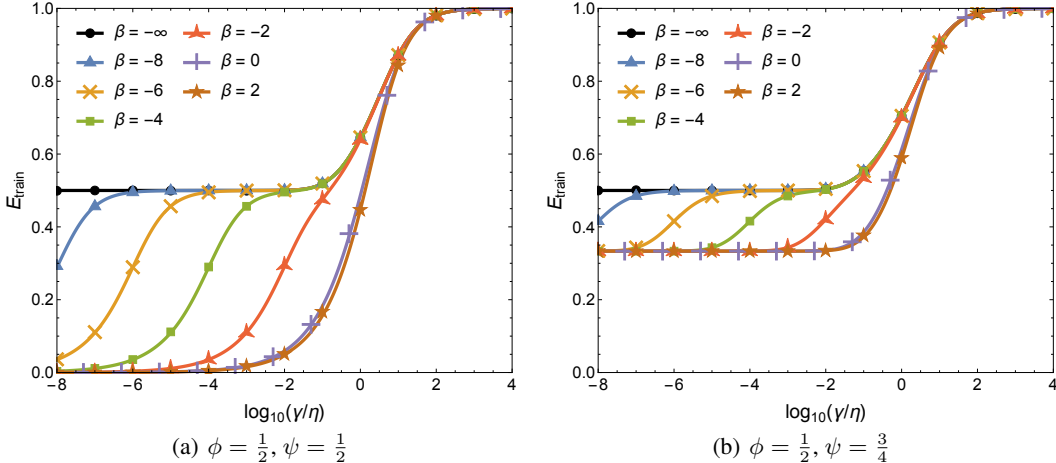

$$\text{(a)}\ \phi = \tfrac{1}{2},\ \psi = \tfrac{1}{2} \qquad\qquad\qquad \text{(b)}\ \phi = \tfrac{1}{2},\ \psi = \tfrac{3}{4}$$

Figure 2: Memorization performance of random feature networks versus ridge regularization parameter $\gamma$. Theoretical curves are solid lines and numerical solutions to eqn. (19) are points. $\beta \equiv \log_{10}(\eta/\zeta - 1)$ distinguishes classes of nonlinearities, with $\beta = -\infty$ corresponding to a linear network. Each numerical simulation is done with a different randomly-chosen function $f$ and the specified $\beta$. The good agreement confirms that no details about $f$ other than $\beta$ are relevant. In (a), there are more random features than data points, allowing for perfect memorization unless the function $f$ is linear, in which case the model is rank constrained. In (b), there are fewer random features than data points, and even the nonlinear models fail to achieve perfect memorization. For a fixed amount of regularization $\gamma$, curves with larger values of $\beta$ (smaller values of $\zeta$) have lower training loss and hence increased memorization capacity.

numerical evidence that eqn. (11) is in fact the correct asymptotic limit. It also shows how quickly the asymptotic behavior sets in, which is useful for interpreting Fig. 1(b), which shows the distance between $\bar{\rho}_1$ and $\rho_{10}$. Observe that if $\zeta = 0$, $\rho_{10}$ approaches $\bar{\rho}_1$ as the network width increases. This provides evidence for the conjecture that the eigenvalues are in fact preserved as they propagate through the network, but only when $\zeta = 0$, since we see the distances level off at some finite value when $\zeta \neq 0$. We also note that small non-zero values of $\zeta$ may not distort the eigenvalues too much.

These observations suggest a new method of tuning the network for fast optimization. Recent work (Pennington et al., 2017) found that inducing dynamical isometry, i.e. equilibrating the singular value distribution of the input-output Jacobian, can greatly speed up training. In our context, by choosing an activation function with $\zeta \approx 0$, we can induce a similar type of isometry, not of the input-output Jacobian, but of the data covariance matrix as it propagates through the network. We conjecture that inducing this additional isometry may lead to further training speed-ups, but we leave further investigation of these ideas to future work.

## 4.2 Asymptotic performance of random feature methods

Consider the ridge-regularized least squares loss function defined by,

$$L(W_2) = \frac{1}{2n_2 m}\|\mathcal{Y} - W_2^T Y\|_F^2 + \gamma\|W_2\|_F^2, \quad Y = f(WX), \tag{19}$$

where $X \in \mathbb{R}^{n_0 \times m}$ is a matrix of $m$ $n_0$-dimensional features, $\mathcal{Y} \in \mathbb{R}^{n_2 \times m}$ is a matrix of regression targets, $W \in \mathbb{R}^{n_1 \times n_0}$ is a matrix of random weights and $W_2 \in \mathbb{R}^{n_1 \times n_2}$ is a matrix of parameters to be learned. The matrix $Y$ is a matrix of random features[1]. The optimal parameters are,

$$W_2^* = \frac{1}{m}YQ\mathcal{Y}^T, \quad Q = \left(\frac{1}{m}Y^TY + \gamma I_m\right)^{-1}. \tag{20}$$

Our problem setup and analysis are similar to that of (Louart et al., 2017), but in contrast to that work, we are interested in the memorization setting in which the network is trained on random input-output pairs. Performance on this task is then a measure of the capacity of the model, or the complexity of the function class it belongs to. In this context, we take the data $X$ and the targets $\mathcal{Y}$ to be independent Gaussian random matrices. From eqns. (19) and (20), the expected training loss is given by,

$$
\begin{aligned}
E_{\text{train}} = \mathbb{E}_{W,X,\mathcal{Y}} \left[ L(W_2^*) \right] &= \mathbb{E}_{W,X,\mathcal{Y}} \left[ \frac{\gamma^2}{m} \operatorname{tr} \mathcal{Y}^T \mathcal{Y} Q^2 \right] \\
&= \mathbb{E}_{W,X} \left[ \frac{\gamma^2}{m} \operatorname{tr} Q^2 \right] \\
&= -\frac{\gamma^2}{m} \frac{\partial}{\partial \gamma} \mathbb{E}_{W,X} \left[ \operatorname{tr} Q \right] .
\end{aligned}
\tag{21}
$$

It is evident from eqn. (5) and the definition of $Q$ that $\mathbb{E}_{W,X} \left[ \operatorname{tr} Q \right]$ is related to $G(-\gamma)$. However, our results from the previous section cannot be used directly because $Q$ contains the trace $Y^T Y$, whereas $G$ was computed with respect to $YY^T$. Thankfully, the two matrices differ only by a finite number of zero eigenvalues. Some simple bookkeeping shows that

$$
\frac{1}{m} \mathbb{E}_{W,X} \left[ \operatorname{tr} Q \right] = \frac{(1 - \phi/\psi)}{\gamma} - \frac{\phi}{\psi} G(-\gamma) .
\tag{22}
$$

From eqn. (11) and its total derivative with respect to $z$, an equation for $G'(z)$ can be obtained by computing the resultant of the two polynomials and eliminating $G(z)$. An equation for $E_{\text{train}}$ follows; see Section 4 of the supplementary material for details. An analysis of this equation shows that it is homogeneous in $\gamma$, $\eta$, and $\zeta$, i.e., for any $\lambda > 0$,

$$
E_{\text{train}}(\gamma, \eta, \zeta) = E_{\text{train}}(\lambda\gamma, \lambda\eta, \lambda\zeta) .
\tag{23}
$$

In fact, this homogeneity is entirely expected from eqn. (19): an increase in the regularization constant $\gamma$ can be compensated by a decrease in scale of $W_2$, which, in turn, can be compensated by increasing the scale of $Y$, which is equivalent to increasing $\eta$ and $\zeta$. Owing to this homogeneity, we are free to choose $\lambda = 1/\eta$. For simplicity, we set $\eta = 1$ and examine the two-variable function $E_{\text{train}}(\gamma, 1, \zeta)$. The behavior when $\gamma = 0$ is a measure of the capacity of the model with no regularization and depends on the value of $\zeta$,

$$
E_{\text{train}}(0, 1, \zeta) =
\begin{cases}
[1 - \phi]_+ & \text{if } \zeta = 1 \text{ and } \psi < 1, \\
[1 - \phi/\psi]_+ & \text{otherwise.}
\end{cases}
\tag{24}
$$

As discussed in Section 3.2, when $\eta = \zeta = 1$, the function $f$ reduces to the identity. With this in mind, the various cases in eqn. (24) are readily understood by considering the effective rank of the random feature matrix $Y$.

In Fig. 2, we compare our theoretical predictions for $E_{\text{train}}$ to numerical simulations of solutions to eqn. (19). The different curves explore various ratios of $\beta \equiv \log_{10}(\eta/\zeta - 1)$ and therefore probe different classes of nonlinearities. For each numerical simulation, we choose a random quintic polynomial $f$ with the specified value of $\beta$ (for details on this choice, see Section 3 of the supplementary material). The excellent agreement between theory and simulations confirms that $E_{\text{train}}$ depends only on $\beta$ and not on any other details of $f$. The black curves correspond to the performance of a linear network. The results show that for $\zeta$ very close to $\eta$, the models are unable to utilize their nonlinearity unless the regularization parameter is very small. Conversely, for $\zeta$ close to zero, the models exploits the nonlinearity very efficiently and absorb large amounts of regularization without a significant drop in performance. This suggests that small $\zeta$ might provide an interesting class of nonlinear functions with enhanced expressive power. See Fig. 3 for some examples of activation functions with this property.

# 5   Conclusions

In this work we studied the Gram matrix $M = \frac{1}{m} Y^T Y$, where $Y = f(WX)$ and $W$ and $X$ are random Gaussian matrices. We derived a quartic polynomial equation satisfied by the trace of the resolvent of $M$, which defines its limiting spectral density. In obtaining this result, we demonstrated

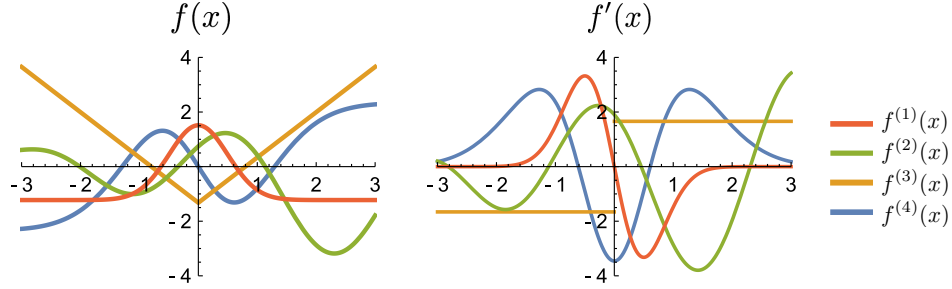

Figure 3: Examples of activation functions and their derivatives for which $\eta = 1$ and $\zeta = 0$. In red, $f^{(1)} = c_1\big(-1 + \sqrt{5}e^{-2x^2}\big)$; in green, $f^{(2)}(x) = c_2\big(\sin(2x) + \cos(3x/2) - 2e^{-2}x - e^{-9/8}\big)$; in orange, $f^{(3)}(x) = c_3\big(|x| - \sqrt{2/\pi}\big)$; and in blue, $f^{(4)}(x) = c_4\big(1 - \frac{4}{\sqrt{3}}e^{-\frac{x^2}{2}}\big)\mathrm{erf}(x)$. If we let $\sigma_w = \sigma_x = 1$, then eqn. (2) is satisfied and $\zeta = 0$ for all cases. We choose the normalization constants $c_i$ so that $\eta = 1$.

that pointwise nonlinearities can be incorporated into a standard method of proof in random matrix theory known as the moments method, thereby opening the door for future study of other nonlinear random matrices appearing in neural networks.

We applied our results to a memorization task in the context of random feature methods and obtained an explicit characterizations of the training error as a function of a ridge regression parameter. The training error depends on the nonlinearity only through two scalar quantities, $\eta$ and $\zeta$, which are certain Gaussian integrals of $f$. We observe that functions with small values of $\zeta$ appear to have increased capacity relative to those with larger values of $\zeta$.

We also make the surprising observation that for $\zeta = 0$, the singular value distribution of $f(WX)$ is the same as the singular value distribution of $X$. In other words, the eigenvalues of the data covariance matrix are constant in distribution when passing through a single nonlinear layer of the network. We conjectured and found numerical evidence that this property actually holds when passing the signal through multiple layers. Therefore, we have identified a class of activation functions that maintains approximate isometry at initialization, which could have important practical consequences for training speed.

Both of our applications suggest that functions with $\zeta \approx 0$ are a potentially interesting class of activation functions. This is a large class of functions, as evidenced in Fig. 3, among which are many types of nonlinearities that have not been thoroughly explored in practical applications. It would be interesting to investigate these nonlinearities in future work.

## Footnotes

[1]We emphasize that we are using an unconvential notation for the random features – we call them $Y$ in order to make contact with the previous sections.

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
