[Supplementary Material]

# Supplemental Material: Nonlinear random matrix theory for deep learning

## 1 Outline of proof of Theorem 1

### 1.1 Polygonal Graphs

Expanding out the powers of $M$ in the equation for moments $\mathbb{E}\left[\frac{1}{n_1} \operatorname{tr} M^k\right]$, we have,

$$\mathbb{E}\left[\frac{1}{n_1} \operatorname{tr} M^k\right] = \frac{1}{n_1}\frac{1}{m^k} \mathbb{E}\left[\sum_{\substack{i_1,\ldots,i_k \in [n_1] \\ \mu_1,\ldots,\mu_k \in [m]}} Y_{i_1\mu_1} Y_{i_2\mu_1} Y_{i_2\mu_2} Y_{i_3\mu_2} \cdots Y_{i_k\mu_k} Y_{i_1\mu_k}\right]. \tag{S1}$$

Notice that this sum can be decomposed based on the pattern of unique $i$ and $\mu$ indices, and, because the elements of $Y$ are i.i.d., the expected value of terms with the same index pattern is the same. Therefore, we are faced with the task of identifying the frequency of each index pattern and the corresponding expected values to leading order in $n_0$ as $n_0 \to \infty$.

To facilitate this analysis, it is useful to introduce a diagrammatic representation of the terms in eqn. (S1). For each term, i.e. each instantiation of indices $i$ and $\mu$ in the sum, we will define a graph.

Consider first any term in which all indices are unique. In this case, we can identify each index with a vertex and each factor $Y_{i_j\mu_j}$ with an edge, and the corresponding graph can be visualized as a $2k$-sided polygon. There is a canonical planar embedding of such a cycle.

More generally, certain indices may be equal in the term. In this case, we can think of the term as corresponding to a polygonal cycle where certain vertices have been identified. The graph now looks like a union of cycles, each joined to another at a common vertex.

Finally, we define *admissible index identifications* as those for which no $i$ index is identified with a $\mu$ index and for which no pairings are crossing (with respect to the canonical embedding). The *admissible graphs* for $k = 3$ are shown in Figure S1, and for $k = 4$ in Figure S2.

**Proposition 1.** *Every admissible graph is a connected outer-planar graph in which all blocks are simple even cycles.*

The proof follows from a simple inductive argument. We will show that these admissible graphs determine the asymptotic (in $n_0$) value of the expectation.

### 1.2 Calculation of Moments

Let $E_G$ denote the expected value of a term in eqn. (S1) corresponding to a graph $G$. We begin with the case where $G$ is a $2k$-cycle. Each $2k$-cycle represents a multi-dimensional integral over the elements of $W$ and $X$. Here we establish a correspondence between these integrals and a lower-dimensional integral whose structure is defined by the adjacency matrix of the graph. For a given $2k$-cycle, the expectation we wish to compute is,

$$
\begin{aligned}
E_{2k} &\equiv \mathbb{E}\left[Y_{i_1\mu_1} Y_{i_2\mu_1} \cdots Y_{i_k\mu_k} Y_{i_1\mu_k}\right] \tag{S2}\\
&= \int f\left(\sum_l W_{i_1 l} X_{l\mu_1}\right) f\left(\sum_l W_{i_2 l} X_{l\mu_1}\right) \cdots f\left(\sum_l W_{i_k l} X_{l\mu_k}\right) f\left(\sum_l W_{i_1 l} X_{l\mu_k}\right) \mathcal{D}W\mathcal{D}X
\end{aligned}
$$

where,

$$\mathcal{D}W = \prod_{i,j=1}^{n_1,n_0} \frac{dW_{ij}}{\sqrt{2\pi\sigma_w^2/n_0}} e^{-\frac{n_0}{2\sigma_w^2}W_{ij}^2} \quad \mathcal{D}X = \prod_{i,\mu=1}^{n_0,d} \frac{dX_{i\mu}}{\sqrt{2\pi\sigma_x^2}} e^{-\frac{1}{2\sigma_x^2}X_{i\mu}^2}, \tag{S3}$$

and $i_1 \neq i_2 \neq ... \neq i_k \neq \mu_1 \neq \mu_2 \neq ... \neq \mu_k$. Next we introduce auxilliary integrals over $z$, which we can do by adding delta function contraints enforcing $Z = WX$. To this end, let $\mathcal{Z}$ denote the set of unique $Y_{i\mu}$ in eqn. (S2). Let $Z \in \mathbb{R}^{n_0 \times d}$ be the matrix whose entries are,

$$Z_{i\mu} = \begin{cases} z_{i\mu} & \text{if } Y_{i\mu} \in \mathcal{Z} \\ 0 & \text{otherwise}. \end{cases} \tag{S4}$$

For each $y \in \mathcal{Z}$ we introduce an auxilliary integral,

$$E_{2k} = \int \prod_{z_{\alpha\beta}\in\mathcal{Z}} \delta(z_{\alpha\beta}-\sum_k W_{\alpha k}X_{k\beta})\, f(z_{i_1\mu_1})f(z_{i_2\mu_1})\cdots f(z_{i_k\mu_k})f(z_{i_1\mu_k})\mathcal{D}z\mathcal{D}W\mathcal{D}X\,, \quad \text{(S5)}$$

where

$$\mathcal{D}z = \prod_{z_{\alpha\beta}\in\mathcal{Z}} dz_{\alpha\beta}\,. \quad \text{(S6)}$$

Next we use a Fourier representation of the Dirac delta function,

$$\delta(x) = \frac{1}{2\pi}\int d\lambda\, e^{i\lambda x}\,, \quad \text{(S7)}$$

for each of the delta functions in eqn. (S5). As above, we define a matrix $\Lambda \in \mathcal{R}^{n_1 \times d}$ whose entries are,

$$\Lambda_{i\mu} = \begin{cases} \lambda_{i\mu} & \text{if } Y_{i\mu} \in \mathcal{Z} \\ 0 & \text{otherwise}\,. \end{cases} \quad \text{(S8)}$$

Then we can write,

$$E_{2k} = \int e^{-i\,\mathrm{tr}\,\Lambda^T(WX-Z)} f(z_{i_1\mu_1})f(z_{i_2\mu_1})\cdots f(z_{i_k\mu_k})f(z_{i_1\mu_k})\,\mathcal{D}\lambda\mathcal{D}z\mathcal{D}W\mathcal{D}X, \quad \text{(S9)}$$

where,

$$\mathcal{D}\lambda = \prod_{\lambda_{\alpha\beta}\in\Lambda} \frac{d\lambda_{\alpha\beta}}{2\pi}\,. \quad \text{(S10)}$$

Note that the integral is bounded so we can use Fubini-Tonelli Theorem to switch integrals and perform the $X$ and $W$ integrals before $\lambda$ and $z$ integrals. We first perform the $X$ integrals,

$$\begin{aligned}
\int \mathcal{D}X\, e^{-i\,\mathrm{tr}\,\Lambda^T WX} &= \prod_{b,c=1}^{d,n_0} \int \frac{dX_{cb}}{\sqrt{2\pi\sigma_x^2}}\,\exp\left[-\frac{1}{2\sigma_x^2}X_{cb}^2 - i\sum_{a=1}^{n_1}\lambda_{ab}W_{ac}X_{cb}\right] \\
&= \exp\left[-\frac{\sigma_x^2}{2}\sum_{a,b,c=1}^{n_1,d,n_0}\left(\lambda_{ab}W_{ac}\right)^2\right] \\
&= e^{-\frac{\sigma_x^2}{2}\,\mathrm{tr}\,\Lambda\Lambda^T WW^T}\,.
\end{aligned} \quad \text{(S11)}$$

Next we perform the $W$ integrals,

$$\begin{aligned}
\int \mathcal{D}W\, e^{-\frac{\sigma_x^2}{2}\,\mathrm{tr}\,\Lambda\Lambda^T WW^T} &= \prod_{i,j=1}^{n_1,n_0}\int \frac{dw_{i,j}}{(2\pi\sigma_w^2/n_0)^{1/2}}\, e^{-\frac{1}{2}\,\mathrm{tr}\left(\frac{n_0}{\sigma_w^2}I_{n_1}+\sigma_x^2\Lambda\Lambda^T\right)WW^T} \\
&= \prod_{j=1}^{n_0}\int \frac{d^{n_1}w_j}{(2\pi\sigma_w^2/n_0)^{n_1/2}}\,\exp\left[-\frac{1}{2}w_j^T\left(\frac{n_0}{\sigma_w^2}I_{n_1}+\sigma_x^2\Lambda\Lambda^T\right)w_j\right] \\
&= \prod_{i=1}^{n_0}\frac{1}{\det|I_{n_1}+\frac{\sigma_w^2\sigma_x^2}{n_0}\Lambda\Lambda^T|^{1/2}} \\
&= \frac{1}{\det|I_{n_1}+\frac{\sigma_w^2\sigma_x^2}{n_0}\Lambda\Lambda^T|^{n_0/2}}\,,
\end{aligned} \quad \text{(S12)}$$

where $w_j \in \mathbb{R}^{n_1}$ is the $j$th column of $W$ and $I_{n_1}$ is the $n_1 \times n_1$ identity matrix. Compiling the results up until now gives,

$$E_{2k} = \int \mathcal{D}\lambda\mathcal{D}z\,\frac{e^{-i\,\mathrm{tr}\,\Lambda Z}}{\det|1+\frac{\sigma_w^2\sigma_x^2}{n_0}\Lambda\Lambda^T|^{n_0/2}}F(z), \quad \text{(S13)}$$

where we have introduced the abbreviation,

$$F(z) = f(z_{i_1\mu_1})f(z_{i_2\mu_1})\cdots f(z_{i_k\mu_k})f(z_{i_1\mu_k}) \quad \text{(S14)}$$

to ease the notation. So far, we have not utilized the fact that $n_0$, $n_1$, and $d$ are large. To proceed, we will use this fact to perform the $\lambda$ integrals in the saddle point approximation, also known as the method of steepest descent. To this end, we write

$$E_{2k} = \int \mathcal{D}\lambda\mathcal{D}z \ \exp\left[-\frac{n_0}{2}\log\det\left|1 + \frac{\sigma_w^2\sigma_x^2}{n_0}\Lambda\Lambda^T\right| - i\operatorname{tr}\Lambda Z\right] F(z) \tag{S15}$$

and observe that the $\lambda$ integrals will be dominated by contributions near where the coefficient of $n_0$ is minimized. It is straightforward to see that the minimizer is $\Lambda = 0$, at which point the phase factor $\operatorname{tr}\Lambda Z$ vanishes. Because the phase factor vanishes at the minimzer, we do not need to worry about the complexity of the integrand, and the approximation becomes equivalent to what is known as Laplace's method. The leading contributions to the integral come from the first non-vanishing terms in the expansion around the minimizer $\Lambda = 0$. To perform this expansion, we use the following identity, valid for small $X$,

$$\log\det|1 + X| = \sum_{j=1}^{\infty} \frac{(-1)^{j+1}}{j}\operatorname{tr}X^j. \tag{S16}$$

Using this expansion, we have,

$$
\begin{aligned}
E_{2k} &= \int \mathcal{D}\lambda\mathcal{D}z \ e^{-\frac{1}{2}\sigma_w^2\sigma_x^2\operatorname{tr}\Lambda\Lambda^T} e^{-\frac{n_0}{2}\sum_{j=2}^{\infty}\frac{(-1)^{j+1}}{j}\operatorname{tr}(\frac{\sigma_w^2\sigma_x^2}{n_0}\Lambda\Lambda^T)^j} e^{-i\operatorname{tr}\Lambda Z} F(z) \\
&= \int \mathcal{D}\tilde{\lambda}\mathcal{D}z \ e^{-\frac{n_0}{2}\operatorname{tr}\tilde{\Lambda}\tilde{\Lambda}^T} e^{-\frac{n_0}{2}\sum_{j=2}^{\infty}\frac{(-1)^{j+1}}{j}\operatorname{tr}(\tilde{\Lambda}\tilde{\Lambda}^T)^j} e^{-i\frac{\sqrt{n_0}}{\sigma_w\sigma_x}\operatorname{tr}\tilde{\Lambda}Z} F(z),
\end{aligned} \tag{S17}
$$

where we have changed integration variables to $\tilde{\lambda}_{ij} = \frac{\sigma_w\sigma_x}{\sqrt{n_0}}\lambda_{ij}$ and

$$\mathcal{D}\tilde{\lambda} = \prod_{\tilde{\lambda}_{\alpha\beta}\in\tilde{\Lambda}} \frac{d\tilde{\lambda}_{\alpha\beta}}{2\pi\sigma_w\sigma_x/\sqrt{n_0}}. \tag{S18}$$

To extract the asymptotic contribution of this integral, we need to understand traces of $\tilde{\Lambda}\tilde{\Lambda}^T$. To this end, we make the following observation.

**Lemma 1.** *Given the matrix $\tilde{\Lambda} = [\tilde{\lambda}_{ij}]$, there exists matrix $A$ such that*

$$\operatorname{tr}(\tilde{\Lambda}\tilde{\Lambda}^T)^k = \frac{1}{2}\operatorname{tr}A^{2k}, \tag{S19}$$

*where $A$ is the weighted adjacency matrix defined by the undirected bigraph with vertex set $V = (I, U)$, where*

$$
\begin{aligned}
I &\equiv \{2i \,|\, \exists\mu \text{ s.t. } y_{i\mu} \in \mathcal{Z}\}, \tag{S20} \\
U &\equiv \{2\mu - 1 \,|\, \exists i \text{ s.t. } y_{i\mu} \in \mathcal{Z}\}, \tag{S21}
\end{aligned}
$$

*and edges,*

$$E \equiv \{\{2\mu - 1, 2i\} \,|\, y_{i\mu} \in \mathcal{Z}\}, \tag{S22}$$

*with weights $w(\{2\mu - 1, 2i\}) = \tilde{\lambda}_{i\mu}$.*

The proof follows by defining an adjacency matrix:

$$A = \begin{pmatrix} 0 & \tilde{\Lambda} \\ \tilde{\Lambda}^T & 0 \end{pmatrix}, \tag{S23}$$

where the $I$ vertices are ordered before $U$ vertices, and observe that the weights agree. Therefore,

$$A^{2k} = \begin{pmatrix} (\tilde{\Lambda}\tilde{\Lambda}^T)^k & 0 \\ 0 & (\tilde{\Lambda}^T\tilde{\Lambda})^k \end{pmatrix}. \tag{S24}$$

Observe that the traces agree as required.

Now suppose that the middle exponential factor appearing in eqn. (S17) is truncated to finite order, $m$,

$$e^{-\frac{n_0}{2}\sum_{j=2}^{m}\frac{(-1)^{j+1}}{j}\operatorname{tr}(\tilde{\Lambda}\tilde{\Lambda}^T)^j}. \tag{S25}$$

Since we are expanding for small $\tilde{\Lambda}$, we can expand the exponential into a polynomial of order $2m$. Any term in this polynomial that does not contain at least one factor $\tilde{\lambda}_{i\mu}$ for each $Y_{i\mu} \in \mathcal{Z}$ will vanish. To see this, denote (any one of) the missing $\tilde{\lambda}_{i\mu}$ as $\tilde{\lambda}$ and the corresponding $z_{i\mu}$ as $z$. Then,

$$
\begin{aligned}
\int dz \int \frac{d\tilde{\lambda}}{2\pi\sigma_w\sigma_x/\sqrt{n_0}} e^{-\frac{n_0}{2}\tilde{\lambda}^2} e^{-i\frac{\sqrt{n_0}}{\sigma_w\sigma_x}\tilde{\lambda}z} f(z) &= \int dz \frac{e^{-\frac{z^2}{2\sigma_w^2\sigma_x^2}}}{\sqrt{2\pi\sigma_w^2\sigma_x^2}} f(z) \\
&= \int \frac{dz}{\sqrt{2\pi}} e^{-\frac{z^2}{2}} f(\sigma_w\sigma_x z) \\
&= 0\,,
\end{aligned}
\tag{S26}
$$

The last line follows from eqn. (2).

The leading contribution to eqn. (S17) comes from the terms in the expansion of eqn. (S25) that have the fewest factors of $\tilde{\lambda}$, while still retaining one factor $\tilde{\lambda}_{i\mu}$ for each $Y_{i\mu}$. Since $\tilde{\lambda} \to 0$, as $n_0 \to \infty$ (the minimizer is $\tilde{\Lambda} = 0$), it follows that if there is a a term with exactly one factor of $\tilde{\lambda}_{i\mu}$ for each $Y_{i\mu}$, it will give the leading contribution. We now argue that there is always such a term, and we compute its coefficient.

Using eqn. (S19), traces of $\operatorname{tr}(\tilde{\Lambda}\tilde{\Lambda}^T)$ are equivalent to traces of $A^2$, where $A$ is the adjacency matrix of the graph defined above. It is well known that the $(u, v)$ entry of the $A^k$ is the sum over weighted walks of length $k$, starting at vertex $u$ and ending at vertex $v$. If there is a cycle of length $k$ in the graph, then the diagonal elements of $A^k$ contain two terms with exactly one factor of $\tilde{\lambda}$ for each edge in the cycle. (There are two terms arising from the clockwise and counter-clockwise walks around the cycle). Therefore, if there is a cycle of length $2k$, the expression $1/2 \operatorname{tr} A^{2k}$ contains a term with one factor of $\tilde{\lambda}$ for each edge in the cycle, with coefficient equal to $2k$.

So, finally, we can write for $k > 1$,

$$
\begin{aligned}
E_{2k} &= \int \mathcal{D}\tilde{\lambda}\mathcal{D}z \; e^{-\frac{n_0}{2}\operatorname{tr}\tilde{\Lambda}\tilde{\Lambda}^T} e^{-\frac{n_0}{2}\sum_{j=2}^{\infty}\frac{(-1)^{j+1}}{j}\operatorname{tr}(\tilde{\Lambda}\tilde{\Lambda}^T)^j} e^{-i\frac{\sqrt{n_0}}{\sigma_w\sigma_x}\operatorname{tr}\tilde{\Lambda}Z} F(z) \\
&\approx (-1)^k n_0 \int \mathcal{D}\tilde{\lambda}\mathcal{D}z \; e^{-\frac{n_0}{2}\operatorname{tr}\tilde{\Lambda}\tilde{\Lambda}^T} e^{-i\frac{\sqrt{n_0}}{\sigma_w\sigma_x}\operatorname{tr}\tilde{\Lambda}Z} \tilde{\lambda}_{i_1\mu_1}\tilde{\lambda}_{i_2\mu_1}\cdots\tilde{\lambda}_{i_k\mu_k}\tilde{\lambda}_{i_1\mu_k} F(z) \\
&= (-1)^k n_0 \left[ \int \frac{d\tilde{\lambda}}{2\pi\sigma_w\sigma_x/\sqrt{n_0}} dz \; e^{-\frac{n_0}{2}\tilde{\lambda}^2} e^{-i\frac{\sqrt{n_0}}{\sigma_w\sigma_x}\tilde{\lambda}z} \tilde{\lambda} f(z) \right]^{2k} \\
&= (-1)^k n_0 \left[ -i \int dz \frac{e^{-\frac{z^2}{2\sigma_w^2\sigma_x^2}}}{\sqrt{2\pi n_0 \sigma_w^2\sigma_x^2}} z f(z) \right]^{2k} \\
&= n_0^{1-k} \left[ \sigma_w\sigma_x \int dz \frac{e^{-z^2/2}}{\sqrt{2\pi}} f'(\sigma_w\sigma_x z) \right]^{2k} \\
&= n_0^{1-k} \zeta^k
\end{aligned}
\tag{S27}
$$

where in the second to last line we have integrated by parts and we have defined,

$$
\zeta \equiv \left[ \sigma_w\sigma_x \int dz \frac{e^{-z^2/2}}{\sqrt{2\pi}} f'(\sigma_w\sigma_x z) \right]^2 .
\tag{S28}
$$

We also note that if $k = 1$, there is no need to expand beyond first order because those integrals will not vanish (as they did in eqn. (S26)). So in this case,

$$
\begin{aligned}
E_2 &\approx \int \mathcal{D}\tilde{\lambda}\mathcal{D}z\; e^{-\frac{n_0}{2}\operatorname{tr}\tilde{\Lambda}\tilde{\Lambda}^T} e^{-i\frac{\sqrt{n_0}}{\sigma_w \sigma_x}\operatorname{tr}\tilde{\Lambda}Z} F(z) \\
&= \left[ \int \frac{d\tilde{\lambda}}{2\pi\sigma_w\sigma_x/\sqrt{n_0}} dz\; e^{-\frac{n_0}{2}\tilde{\lambda}^2} e^{-i\frac{\sqrt{n_0}}{\sigma_w\sigma_x}\tilde{\lambda}z} f(z) \right]^{2k} \\
&= \left[ \int dz\; \frac{e^{-\frac{z^2}{2\sigma_w^2\sigma_x^2}}}{\sqrt{2\pi n_0 \sigma_w^2 \sigma_x^2}} f(z) \right]^{2k} \\
&= \int dz\; \frac{e^{-z^2/2}}{\sqrt{2\pi}} f(\sigma_w\sigma_x z)^2 \\
&\equiv \eta\,.
\end{aligned}
\tag{S29}
$$

The quantities $\eta$ and $\zeta$ are important and will be used to obtain an expression for $G$.

The above was the simplest case, a $2k$ cycle. For any admissible graph $G$, we can view it as a tree over blocks, each block being a (even) cycle. If $G$ has $2k$ edges then one can write the integral above as a product of integrals over cyclic blocks. In this case, each block contributes a factor of $n_0$ to the integral, and if there are $c$ cyclic blocks, with $k_1$ blocks of size 1 and $k_0$ blocks of size greater than 1, the resulting expression for the integral has value $n_0^{k_0-k}\zeta^{k_0}\eta^{k_1}$. Since $k = k_0 = 1$ for a 2-cycle, we have the following proposition.

**Proposition 2.** *Given an admissible graph $G$ with $c$ cyclic blocks, $b$ blocks of size 1, and $2k$ edges, $E_G$ grows as $n_0^{c-k} \cdot \eta^b \zeta^{c-b}$.*

### 1.2.1 Non-Admissible Graphs

Finally, we note that the terms contributing to the admissible graphs determine the asymptotic value of the expectation. The number of terms (and therefore graphs) with $k$ indices and $c$ identifications is $\Theta(n_0^{2k-c})$. Although the fraction of non-admissible graphs with $c$ identifications is far larger than that of admissible graphs as a function of $k$, the leading term for the integrals corresponding to latter grow as $n_0^{c-k}$, while the leading terms for the former grow at most as $n_0^{c-1-k}$. The underlying reason for this subleading scaling is that any partition of a non-admissible graph into $c$ blocks, where no two blocks have an edge in-between, requires $c$ index identifications in the original $2k$-polygon, as opposed to $c-1$ identifications for an admissible graph. Therefore, we may restrict our attention only to admissible graphs in order to complete the asymptotic evaluation of $G$.

### 1.3 Generating function

Let $\tilde{p}(k, v_i, v_\mu, b)$ denote the number of admissible graphs with $2k$ edges, $v_i$ $i$-type vertex identifications, $v_\mu$ $\mu$-type vertex identifications, and $b$ cycles of size 1. Similarly, let $p(k, v_i, v_\mu, b)$ denote the same quantity modulo permutations of the vertices. Then, combining the definition of $G(z)$ (eqn. (7)) and Proposition 2, we have,

$$
\begin{aligned}
G(z) &\simeq \frac{1}{z} + \sum_{k=1}^{\infty}\sum_{v_i,v_\mu=0}^{k}\sum_{b=0}^{v_i+v_\mu+1} \binom{n_1}{k-v_i}\binom{m}{k-v_\mu} \frac{\tilde{p}(k,v_i,v_\mu,b)}{z^{k+1}} \frac{n_0^{v+v_\mu+1-k}}{n_1 m^k}\eta^b \zeta^{v_i+v_\mu+1-b} \\
&\simeq \frac{1}{z} + \sum_{k=1}^{\infty}\frac{1}{z^{k+1}}\sum_{v_i,v_\mu=0}^{k}\sum_{b=0}^{v+v_\mu+1} p(k,v_i,v_\mu,b)\eta^b \zeta^{v_i+v_\mu+1-b}\phi^{v_i}\psi^{v_\mu} \\
&\simeq \frac{1-\psi}{z} + \frac{\psi}{z}\sum_{k=0}^{\infty}\frac{1}{z^k\psi^k}P(k),
\end{aligned}
\tag{S30}
$$

where we have defined,

$$P(k) = \sum_{v_i, v_\mu=0}^{k} \sum_{b=0}^{v_i+v\mu+1} p(k, v_i, v_\mu, b) \eta^b \zeta^{v_i+v\mu+1-b} \phi^{v_i} \psi^{v_\mu} .$$ 

(S31)

Let $P(t) = \sum_k P(k)t^k$ be a generating function. Let $2k$ refer to the size of the cycle containing vertex 1. Summing over all possible values of $k$ yields the following recurrence relation,

$$\begin{aligned} P &= 1 + tP_\phi P_\psi \eta + \sum_{k=2}^{\infty} (P_\phi P_\psi \zeta t)^k \\ &= 1 + (\eta - \zeta)tP_\phi P_\psi + \frac{P_\phi P_\psi t\zeta}{1 - P_\phi P_\psi t\zeta} . \end{aligned}$$

(S32)

Note that if vertex 1 is inside a bubble, we get a factor of $\eta$ instead of $\zeta$, which is why that term is treated separately. The auxilliary generating functions $P_\phi$ and $P_\psi$ correspond to the generating functions of graphs with an extra factor $\phi$ or $\psi$ respectively, i.e.

$$P_\phi = 1 + (P-1)\phi \quad P_\psi = 1 + (P-1)\psi ,$$

(S33)

which arises from making a $i$-type or $\mu$-type vertex identifications. Accounting for the relation between $G$ and $P$ in eqn. (S30) yields,

$$G(z) = \frac{\psi}{z} P\left(\frac{1}{z\psi}\right) + \frac{1-\psi}{z} .$$

(S34)

Hence, we have completed our outline of the proof of Theorem 1.

## 2 Example Graphs

Figure S1: Admissible graphs for $k = 3$

Figure S2: Admissible topologies for $k = 4$

## 3 Hermite expansion

Any function with finite Gaussian moments can be expanded in a basis of Hermite polynomials. Defining

$$H_n(x) = (-1)^n e^{\frac{x^2}{2}} \frac{\partial^n}{\partial x^n} e^{-\frac{x^2}{2}} \tag{S35}$$

we can write,

$$f(x) = \sum_{n=0}^{\infty} \frac{f_n}{\sqrt{n!}} H_n(x), \tag{S36}$$

for some constants $f_n$. Owing the orthogonality of the Hermite polynomials, this representation is useful for evaluating Gaussian integrals. In paticular, the condition that $f$ be centered is equivalent the vanishing of $f_0$,

$$\begin{aligned} 0 &= \int_{-\infty}^{\infty} dx \frac{e^{-x^2/2}}{\sqrt{2\pi}} f(x) \\ &= f_0. \end{aligned} \tag{S37}$$

The constants $\eta$ and $\zeta$ are also easily expressed in terms of the coefficients,

$$\begin{aligned} \eta &= \int_{-\infty}^{\infty} dx \frac{e^{-x^2/2}}{\sqrt{2\pi}} f(x)^2 \\ &= \sum_{n=0}^{\infty} f_n^2, \end{aligned} \tag{S38}$$

and,

$$\begin{aligned} \zeta &= \left[ \int_{-\infty}^{\infty} dx \frac{e^{-x^2/2}}{\sqrt{2\pi}} f'(x) \right]^2 \\ &= f_1^2, \end{aligned} \tag{S39}$$

which together imply that $\eta \geq \zeta$. Equality holds when $f_{i>1} = 0$, in which case,

$$f(x) = f_1 H_1(x) = f_1 x \tag{S40}$$

i.e. when $f$ is a linear function.

The Hermite representation also suggests a convenient way to randomly sample functions with specified values of $\eta$ and $\zeta$. First choose $f_1 = \sqrt{\zeta}$, and then enforce the constraint,

$$\eta - 1 = \sum_{n=2}^{N} f_n^2, \tag{S41}$$

where we have truncated the representation to some finite order $N$. Random values of $f_n$ satisfying this relation are simple to obtain since they all live on the sphere of radius $\sqrt{\eta - 1}$.

## 4 Equations for Stieltjes transform

From eqn. (11), straightforward algebra shows that $G$ satisfies,

$$\sum_{i=0}^{4} a_i G^i = 0 \,, \tag{S42}$$

where,

$$
\begin{aligned}
a_0 &= -\psi^3 \,, \\
a_1 &= \psi(\zeta(\psi - \phi) + \psi(\eta(\phi - \psi) + \psi z)) \\
a_2 &= -\zeta^2(\phi - \psi)^2 + \zeta \left(\eta(\phi - \psi)^2 + \psi z(2\phi - \psi)\right) - \eta\psi^2 z\phi \\
a_3 &= \zeta(-z)\phi(2\zeta\psi - 2\zeta\phi - 2\eta\psi + 2\eta\phi + \psi z) \\
a_4 &= \zeta z^2 \phi^2 (\eta - \zeta) \,.
\end{aligned}
\tag{S43}
$$

The total derivative of this equation with respect to $z$ is,

$$\sum_{i=1}^{4} a_i' G^i + G' \sum_{i=0}^{3} b_i G^i = 0 \,, \tag{S44}$$

where,

$$
\begin{aligned}
a_1' &= \psi^3 \,, \\
a_2' &= -\psi(\zeta(\psi - 2\phi) + \eta\psi\phi) \,, \\
a_3' &= -2\zeta\phi(\zeta(\psi - \phi) + \eta(\phi - \psi) + \psi z) \,, \\
a_4' &= 2\zeta z\phi^2(\eta - \zeta) \,, \\
b_0 &= \psi(\zeta(\psi - \phi) + \psi(\eta(\phi - \psi) + \psi z)) \\
b_1 &= 2\eta\left(\zeta(\phi - \psi)^2 - \psi^2 z\phi\right) - 2\zeta\left(\zeta(\phi - \psi)^2 + \psi z(\psi - 2\phi)\right) \\
b_2 &= -3\zeta z\phi(2\zeta\psi - 2\zeta\phi - 2\eta\psi + 2\eta\phi + \psi z) \\
b_3 &= 4\zeta z^2 \phi^2 (\eta - \zeta) \,.
\end{aligned}
\tag{S45}
$$

To eliminate $G$ from eqs. (S43) and (S45), we compute the resultant of the two polynomials, which produces a quartic polynomial in $G'$. Using eqns. (21) and (22) to change variables to $E_{\text{train}}$, we can derive the following equation satisfied by $E_{\text{train}}$,

$$\sum_{i=0}^{4} \sum_{j=0}^{6} c_{i,j} \gamma^j E_{\text{train}}^i = 0 \,, \tag{S46}$$

where the $c_{i,j}$ are given below. Notice that $\eta = \zeta$ is a degenerate case since $a_4 = b_3 = 0$ and the resultant must be computed separately. We find,

$$\sum_{i=0}^{3} \sum_{j=0}^{4} d_{i,j} \gamma^j E_{\text{train}}^i \big|_{\eta=\zeta} = 0 \,, \tag{S47}$$

where the $d_{i,j}$ are given below. By inspection we find that

$$c_{i,j}(\lambda\eta, \lambda\zeta) = \lambda^{8-j} c_{i,j}(\eta, \zeta) \quad \text{and} \quad d_{i,j}(\lambda\zeta) = \lambda^{4-j} d_{i,j}(\zeta) \,, \tag{S48}$$

which establishes the homogeneity of $E_{\text{train}}$ in $\gamma$, $\eta$, and $\zeta$. From the coefficients $c_{i,0}$ we can read off the quartic equation satisfied by $E_{\text{train}}$ when $\gamma = 0$ and $\eta \neq \zeta$. It has two double roots at,

$$E_{\text{train}}|_{\gamma=0} = 0 \quad \text{and} \quad E_{\text{train}}|_{\gamma=0} = 1 - \phi/\psi \,. \tag{S49}$$

In accordance with the condition that $G \to 1/z$ as $z \to \infty$, the first root is chosen if $\psi < \phi$ and the second root chosen otherwise.

If $\eta = \zeta$, then the coefficients $d_{i,0}$ define a cubic equation for $E_{\text{train}}$ that has three distinct roots,

$$E_{\text{train}}|_{\gamma=0,\eta=\zeta} = 0 \,, \quad E_{\text{train}}|_{\gamma=0,\eta=\zeta} = 1 - \phi \,, \quad \text{and} \quad E_{\text{train}}|_{\gamma=0,\eta=\zeta} = 1 - \phi/\psi \,. \tag{S50}$$

In this case, the first root is chosen when $\phi > \max(\psi, 1)$, the second root is chosen when $\phi, \psi < 1$, and the third root chosen otherwise.

Finally we give the coefficients $c_{i,j}$,

$$c_{0,0} = 0, \quad c_{0,1} = 0, \quad c_{0,2} = 0, \quad c_{0,3} = 0,$$
$$c_{1,0} = 0, \quad c_{1,1} = 0, \quad c_{3,6} = 0, \quad c_{4,6} = 0,$$

and,

$$c_{0,4} = \psi^6 \phi^3 \big(\zeta^2(4\psi - 1) - 2\zeta\eta\psi - \eta^2\psi^2\big)\big(\zeta^2\big((\psi - 1)\psi + \phi^2 + 2\psi\phi\big) - 2\zeta\eta\psi\phi - \eta^2\psi\phi^2\big)$$

$$c_{0,5} = 2\zeta\psi^8\phi^3\big(\zeta^2(-\psi + \phi + 1) + \zeta\eta\big(-\psi^2 + \psi - 3\psi\phi + \phi\big) + \eta^2\psi\phi\big)$$

$$c_{0,6} = -\zeta^2(\psi - 1)\psi^9\phi^3$$

$$c_{1,2} = \psi^4\phi(\phi - \psi)^3\big(\zeta^2(4\psi - 1) - 2\zeta\eta\psi - \eta^2\psi^2\big)\big(\zeta^2(4\phi - 1) - 2\zeta\eta\phi - \eta^2\phi^2\big)\big(\zeta^2(\psi + \phi - 1) - \eta^2\psi\phi\big)$$

$$c_{1,3} = -2\psi^5\phi(\phi - \psi)\big(\zeta^5\big(-(\psi - 1)\psi^2 - \phi^3 + (-32\psi^2 + 9\psi + 1)\phi^2 + \psi(9\psi - 4)\phi\big) +$$
$$\zeta^4\eta\big(-(\psi - 1)\psi^3 + (4\psi - 1)\phi^4 + (12\psi^2 - 8\psi + 1)\phi^3 + 2\psi(6\psi^2 + 17\psi - 2)\phi^2 + 4\psi^2(\psi^2 - 2\psi - 1)\phi\big) - \zeta^3\eta^2\psi\phi\big((\psi - 2)\psi^2 + \phi^3 + (7\psi - 2)\phi^2 + \psi(7\psi + 8)\phi\big) +$$
$$2\zeta^2\eta^3\psi\phi\big(\psi^3 + (1 - 4\psi)\phi^3 - 4\psi^3\phi\big) + 3\zeta\eta^4\psi^2\phi^2\big(\psi^2 + \phi^2\big) +$$
$$\eta^5\psi^3\phi^3(\psi + \phi)\big)$$

$$c_{1,4} = \psi^6(-\phi)(\phi - \psi)\big(\zeta^4\big(-(\psi - 1)\psi^2 + (4\psi - 1)\phi^3 + (-16\psi^2 + \psi + 1)\phi^2 + \psi(4\psi^2 - \psi - 9)\phi\big) + 2\zeta^3\eta\psi\phi\big((\psi - 1)\psi + \phi^2 + (12\psi - 1)\phi\big) + 2\zeta^2\eta^2\psi\phi\big(3\psi^2 + (3 - 9\psi)\phi^2 +$$
$$(\psi - 8\psi^2)\phi\big) + 6\zeta\eta^3\psi^2\phi^2(\psi + \phi) + \eta^4\psi^3\phi^3\big)$$

$$c_{1,5} = 2\zeta\psi^8\phi^2\big(\zeta^2\big((\psi - 1)\psi - \phi^2 + 2\psi\phi + \phi\big) + 2\zeta\eta\big(\psi^2 + (2\psi - 1)\phi^2 - \psi^2\phi\big) +$$
$$\eta^2\psi\phi(\psi - \phi)\big)$$

$$c_{1,6} = \zeta^2\psi^9\phi^2\big(\psi + (\psi - 1)\phi\big)$$

$$c_{2,0} = \zeta^2\psi^2(\zeta - \eta)^2(\phi - \psi)^6\big(\zeta^2(4\psi - 1) - 2\zeta\eta\psi - \eta^2\psi^2\big)\big(\zeta^2(4\phi - 1) - 2\zeta\eta\phi - \eta^2\phi^2\big)$$

$$c_{2,1} = -2\zeta\psi^3(\zeta - \eta)(\phi - \psi)^4\big(\zeta^5\big(-5\psi^2 + \psi + (16\psi - 5)\phi^2 + (16\psi^2 - 6\psi + 1)\phi\big) +$$
$$\zeta^4\eta\big(-(\psi - 3)\psi^2 + (4\psi - 1)\phi^3 + (-40\psi^2 - 7\psi + 3)\phi^2 + \psi^2(4\psi - 7)\phi\big) + \zeta^3\eta^2\big(2\psi^3 + (2 - 9\psi)\phi^3 + 34\psi^2\phi^2 - 9\psi^3\phi\big) + \zeta^2\eta^3\psi\phi\big(\psi^2 + (8\psi + 1)\phi^2 + 2\psi(4\psi - 3)\phi\big) -$$
$$3\zeta\eta^4\psi^2\phi^2(\psi + \phi) - 2\eta^5\psi^3\phi^3\big)$$

$$c_{2,2} = \psi^4\big(-(\phi - \psi)^2\big)\big(\zeta^6\big(-\psi^2(\psi^2 - 8\psi + 1) + (4\psi - 1)\phi^4 + (-148\psi^2 + 22\psi + 8)\phi^3 -$$
$$\big(148\psi^3 - 118\psi^2 + 21\psi + 1\big)\phi^2 + \psi\big(4\psi^3 + 22\psi^2 - 21\psi + 3\big)\phi\big) - 2\zeta^5\eta\big(-3(\psi - 1)\psi^3 + (11\psi - 3)\phi^4 + (-147\psi^2 + 24\psi + 3)\phi^3 + \psi(-147\psi^2 + 66\psi - 8)\phi^2 + \psi^2(11\psi^2 + 24\psi - 8)\phi\big) +$$
$$\zeta^4\eta^2\big(-6\psi^4 + (66\psi^2 + 9\psi - 6)\phi^4 + \psi(28\psi^2 - 199\psi + 27)\phi^3 + \psi^2(66\psi^2 - 199\psi + 29)\phi^2 +$$
$$9\psi^3(\psi + 3)\phi\big) + 2\zeta^3\eta^3\psi\phi\big(5\psi^3 + (5 - 44\psi)\phi^3 + (21 - 20\psi)\psi\phi^2 + (21 - 44\psi)\psi^2\phi\big) +$$
$$\zeta^2\eta^4\psi^2\phi^2\big(24\psi^2 + (24 - 13\psi)\phi^2 + (23 - 13\psi)\psi\phi\big) + 10\zeta\eta^5\psi^3\phi^3(\psi + \phi) +$$
$$\eta^6\psi^4\phi^4\big)$$

$$c_{2,3} = 2\psi^5\big(\zeta^5\big(-(\psi-1)\psi^4 + (3\psi-1)\phi^5 + \big(-36\psi^2 + 19\psi + 1\big)\phi^4 + \psi\big(98\psi^2 - 26\psi - 7\big)\phi^3 -$$
$$2\psi^2\big(18\psi^2 + 13\psi - 7\big)\phi^2 + \psi^3\big(3\psi^2 + 19\psi - 7\big)\phi\big) + \zeta^4\eta\big(2\psi^5 + \big(-40\psi^2 + 5\psi + 2\big)\phi^5 +$$
$$\psi\big(24\psi^2 + 54\psi - 19\big)\phi^4 + 2\psi^2\big(12\psi^2 - 67\psi + 10\big)\phi^3 + 2\psi^3\big(-20\psi^2 + 27\psi + 10\big)\phi^2 + \psi^4(5\psi -$$
$$19)\phi\big) + \zeta^3\eta^2\psi\phi\big(-11\psi^4 + (50\psi - 11)\phi^4 - 2\psi(21\psi + 1)\phi^3 - 6\psi^2(7\psi - 5)\phi^2 + 2\psi^3(25\psi -$$
$$1)\phi\big) + 2\zeta^2\eta^3\psi^2\phi^2\big(-7\psi^3 + (5\psi - 7)\phi^3 - 2(\psi - 3)\psi\phi^2 + \psi^2(5\psi + 6)\phi\big) +$$
$$\zeta\eta^4\psi^3\phi^3\big(-5\psi^2 - 5\phi^2 + 4\psi\phi\big) - \eta^5\psi^4\phi^4(\psi + \phi)\big)$$

$$c_{2,4} = \psi^6\big(\zeta^4\big(\psi^4 + \big(-31\psi^2 + 7\psi + 1\big)\phi^4 + \psi\big(70\psi^2 - 6\psi - 13\big)\phi^3 + \psi^2\big(-31\psi^2 - 6\psi +$$
$$31\big)\phi^2 + \psi^3(7\psi - 13)\phi\big) + 2\zeta^3\eta\psi\phi\big(-8\psi^3 + (17\psi - 8)\phi^3 + 3(3 - 16\psi)\psi\phi^2 + \psi^2(17\psi +$$
$$9)\phi\big) + \zeta^2\eta^2\psi^2\phi^2\big(-14\psi^2 + (17\psi - 14)\phi^2 + \psi(17\psi + 14)\phi\big) - 6\zeta\eta^3\psi^3\phi^3(\psi + \phi) -$$
$$\eta^4\psi^4\phi^4\big)$$

$$c_{2,5} = -2\zeta\psi^8\phi\big(\zeta^2\big(2\psi^2 + (\psi + 2)\phi^2 + (\psi - 5)\psi\phi\big) + \zeta\eta\psi\phi(2\psi + (2 - 3\psi)\phi) +$$
$$\eta^2\psi^2\phi^2\big)$$

$$c_{2,6} = -\zeta^2\psi^{10}\phi^2$$

$$c_{3,0} = 2\zeta^2\psi^3(\zeta - \eta)^2(\phi - \psi)^5\big(\zeta^2(4\psi - 1) - 2\zeta\eta\psi - \eta^2\psi^2\big)\big(\zeta^2(4\phi - 1) -$$
$$2\zeta\eta\phi - \eta^2\phi^2\big)$$

$$c_{3,1} = -4\zeta\psi^4(\zeta - \eta)(\phi - \psi)^3\big(\zeta^5\big(-5\psi^2 + \psi + (16\psi - 5)\phi^2 + \big(16\psi^2 - 6\psi + 1\big)\phi\big) +$$
$$\zeta^4\eta\big(-(\psi - 3)\psi^2 + (4\psi - 1)\phi^3 + \big(-40\psi^2 - 7\psi + 3\big)\phi^2 + \psi^2(4\psi - 7)\phi\big) + \zeta^3\eta^2\big(2\psi^3 + (2 -$$
$$9\psi)\phi^3 + 34\psi^2\phi^2 - 9\psi^3\phi\big) + \zeta^2\eta^3\psi\phi\big(\psi^2 + (8\psi + 1)\phi^2 + 2\psi(4\psi - 3)\phi\big) -$$
$$3\zeta\eta^4\psi^2\phi^2(\psi + \phi) - 2\eta^5\psi^3\phi^3\big)$$

$$c_{3,2} = -2\zeta\psi^5(\phi - \psi)\big(\zeta^5\big(-\psi^2\big(\psi^2 - 8\psi + 1\big) + (4\psi - 1)\phi^4 + \big(-132\psi^2 + 18\psi + 8\big)\phi^3 -$$
$$\big(132\psi^3 - 94\psi^2 + 16\psi + 1\big)\phi^2 + 2\psi\big(2\psi^3 + 9\psi^2 - 8\psi + 1\big)\phi\big) - 2\zeta^4\eta\big(-3(\psi - 1)\psi^3 + (11\psi -$$
$$3)\phi^4 + \big(-139\psi^2 + 23\psi + 3\big)\phi^3 + \psi\big(-139\psi^2 + 56\psi - 7\big)\phi^2 + \psi^2\big(11\psi^2 + 23\psi - 7\big)\phi\big) +$$
$$2\zeta^3\eta^2\big(-3\psi^4 + \big(31\psi^2 + 5\psi - 3\big)\phi^4 + \psi\big(2\psi^2 - 93\psi + 13\big)\phi^3 + \psi^2\big(31\psi^2 - 93\psi + 12\big)\phi^2 +$$
$$\psi^3(5\psi + 13)\phi\big) + 2\zeta^2\eta^3\psi\phi\big(5\psi^3 + (5 - 43\psi)\phi^3 + (19 - 10\psi)\psi\phi^2 + (19 - 43\psi)\psi^2\phi\big) +$$
$$\zeta\eta^4\psi^2\phi^2\big(23\psi^2 + (23 - 8\psi)\phi^2 + 2(9 - 4\psi)\psi\phi\big) + 8\eta^5\psi^3\phi^3(\psi + \phi)\big)$$

$$c_{3,3} = 4\zeta\psi^6(\phi - \psi)\big(\zeta^4\big(-(\psi - 1)\psi^2 + (3\psi - 1)\phi^3 + \big(-30\psi^2 + 16\psi + 1\big)\phi^2 + \psi\big(3\psi^2 + 16\psi -$$
$$4\big)\phi\big) + 2\zeta^3\eta\big(\psi^3 + \big(-18\psi^2 + 2\psi + 1\big)\phi^3 + \psi\big(-18\psi^2 + 27\psi - 7\big)\phi^2 + \psi^2(2\psi - 7)\phi\big) +$$
$$\zeta^2\eta^2\psi\phi\big(-11\psi^2 + (49\psi - 11)\phi^2 + \psi(49\psi - 22)\phi\big) + 2\zeta\eta^3\psi^2\phi^2((\psi - 6)\phi - 6\psi) -$$
$$2\eta^4\psi^3\phi^3\big)$$

$$c_{3,4} = -2\zeta^2\psi^7(\phi - \psi)\big(\zeta^2\big(-\psi^2 + \big(27\psi^2 - 6\psi - 1\big)\phi^2 + 2(5 - 3\psi)\psi\phi\big) -$$
$$4\zeta\eta\psi\phi((9\psi - 4)\phi - 4\psi) + 8\eta^2\psi^2\phi^2\big)$$

$$c_{3,5} = 8\zeta^3\psi^9\phi(\psi - \phi)$$

$$c_{4,0} = \zeta^2\psi^4(\zeta-\eta)^2(\phi-\psi)^4\big(\zeta^2(4\psi-1)-2\zeta\eta\psi-\eta^2\psi^2\big)\big(\zeta^2(4\phi-1)-$$
$$2\zeta\eta\phi-\eta^2\phi^2\big)$$

$$c_{4,1} = -2\zeta\psi^5(\zeta-\eta)(\phi-\psi)^2\big(\zeta^5\big(-5\psi^2+\psi+(16\psi-5)\phi^2+\big(16\psi^2-6\psi+1\big)\phi\big)+$$
$$\zeta^4\eta\big(-(\psi-3)\psi^2+(4\psi-1)\phi^3+\big(-40\psi^2-7\psi+3\big)\phi^2+\psi^2(4\psi-7)\phi\big)+\zeta^3\eta^2\big(2\psi^3+(2-$$
$$9\psi)\phi^3+34\psi^2\phi^2-9\psi^3\phi\big)+\zeta^2\eta^3\psi\phi\big(\psi^2+(8\psi+1)\phi^2+2\psi(4\psi-3)\phi\big)-$$
$$3\zeta\eta^4\psi^2\phi^2(\psi+\phi)-2\eta^5\psi^3\phi^3\big)$$

$$c_{4,2} = -\zeta\psi^6\big(\zeta^5\big(-\psi^2\big(\psi^2-8\psi+1\big)+(4\psi-1)\phi^4+\big(-132\psi^2+18\psi+8\big)\phi^3-\big(132\psi^3-94\psi^2$$
$$+16\psi+1\big)\phi^2+2\psi\big(2\psi^3+9\psi^2-8\psi+1\big)\phi\big)-2\zeta^4\eta\big(-3(\psi-1)\psi^3+(11\psi-3)\phi^4+\big(-139\psi^2+$$
$$23\psi+3\big)\phi^3+\psi\big(-139\psi^2+56\psi-7\big)\phi^2+\psi^2\big(11\psi^2+23\psi-7\big)\phi\big)+2\zeta^3\eta^2\big(-3\psi^4+$$
$$\big(31\psi^2+5\psi-3\big)\phi^4+\psi\big(2\psi^2-93\psi+13\big)\phi^3+\psi^2\big(31\psi^2-93\psi+12\big)\phi^2+\psi^3(5\psi+13)\phi\big)+$$
$$2\zeta^2\eta^3\psi\phi\big(5\psi^3+(5-43\psi)\phi^3+(19-10\psi)\psi\phi^2+(19-43\psi)\psi^2\phi\big)+$$
$$\zeta\eta^4\psi^2\phi^2\big(23\psi^2+(23-8\psi)\phi^2+2(9-4\psi)\psi\phi\big)+8\eta^5\psi^3\phi^3(\psi+\phi)\big)$$

$$c_{4,3} = 2\zeta\psi^7\big(\zeta^4\big(-(\psi-1)\psi^2+(3\psi-1)\phi^3+\big(-30\psi^2+16\psi+1\big)\phi^2+\psi\big(3\psi^2+16\psi-$$
$$4\big)\phi\big)+2\zeta^3\eta\big(\psi^3+\big(-18\psi^2+2\psi+1\big)\phi^3+\psi\big(-18\psi^2+27\psi-7\big)\phi^2+\psi^2(2\psi-7)\phi\big)+$$
$$\zeta^2\eta^2\psi\phi\big(-11\psi^2+(49\psi-11)\phi^2+\psi(49\psi-22)\phi\big)+2\zeta\eta^3\psi^2\phi^2((\psi-6)\phi-6\psi)-$$
$$2\eta^4\psi^3\phi^3\big)$$

$$c_{4,4} = \zeta^2\psi^8\big(\zeta^2\big(\psi^2+\big(-27\psi^2+6\psi+1\big)\phi^2+2\psi(3\psi-5)\phi\big)+4\zeta\eta\psi\phi((9\psi-4)\phi-$$
$$4\psi)-8\eta^2\psi^2\phi^2\big)$$

$$c_{4,5} = -4\zeta^3\psi^{10}\phi$$

And the coefficients $d_{i,j}$ read,
$$d_{0,0} = 0, \quad d_{0,1} = 0, \quad d_{2,4} = 0, \quad d_{3,4} = 0\,,$$
and,
$$d_{0,2} = -\zeta^2(\psi-1)^2\psi^2\phi^2\big(\phi^2-\psi\big)$$
$$d_{0,3} = 2\zeta\psi^4\phi^2(\psi+2\phi+1)$$
$$d_{0,4} = \psi^5\phi^2$$

$$d_{1,0} = \zeta^4(\psi-1)^2(\phi-1)^3(\phi-\psi)^3$$
$$d_{1,1} = 2\zeta^3\psi(\phi-1)\big(-\psi^3(\psi+1)+\big(\psi^2-4\psi+1\big)\phi^4+\big(6\psi^2+\psi+1\big)\phi^3-\psi\big(\psi^3+6\psi^2+$$
$$5\big)\phi^2+\psi^2\big(4\psi^2-\psi+5\big)\phi\big)$$
$$d_{1,2} = \zeta^2\psi^2\big(\psi^3+\big(\psi^2-11\psi+1\big)\phi^4-\big(\psi^3+1\big)\phi^3+2\psi\big(5\psi^2+6\psi+5\big)\phi^2-$$
$$11\psi^2(\psi+1)\phi\big)$$
$$d_{1,3} = 2\zeta\psi^4\phi\big(-2\psi-3\phi^2+\psi\phi+\phi\big)$$
$$d_{1,4} = \psi^5\big(-\phi^2\big)$$

$$d_{2,0} = \zeta^4(\psi-1)^2(\phi-1)^2(\phi-\psi)^2(-2\psi+\psi\phi+\phi)$$
$$d_{2,1} = 2\zeta^3\psi\big(-2\psi^3(\psi+1)+\big(\psi^3-3\psi^2-3\psi+1\big)\phi^4+\big(\psi^4+\psi^3+12\psi^2+\psi+1\big)\phi^3-$$
$$6\psi\big(\psi^3+\psi^2+\psi+1\big)\phi^2+\psi^2\big(9\psi^2-2\psi+9\big)\phi\big)$$
$$d_{2,2} = \zeta^2\psi^2\big(-2\psi^3+\big(\psi^3-9\psi^2-9\psi+1\big)\phi^3-12\big(\psi^3+\psi\big)\phi^2+21\psi^2(\psi+1)\phi\big)$$
$$d_{2,3} = -4\zeta\psi^4\phi(-2\psi+\psi\phi+\phi)$$

$$d_{3,0} = \zeta^4(\psi-1)^2\psi(\phi-1)^2(\phi-\psi)^2$$

$$d_{3,1} = 2\zeta^3\psi^2\big(\psi^2(\psi+1) + \big(\psi^2-4\psi+1\big)\phi^3 + \big(\psi^3+2\psi^2+2\psi+1\big)\phi^2 + 2\psi\big(-2\psi^2+\psi-2\big)\phi\big)$$

$$d_{3,2} = \zeta^2\psi^3\big(\psi^2 + \big(\psi^2-10\psi+1\big)\phi^2 - 10\psi(\psi+1)\phi\big)$$

$$d_{3,3} = -4\zeta\psi^5\phi$$