[Reviews · NeurIPS 2017]

Reviewer 1



This looks like a good theoretical contribution and an interesting direction in the theory of deep learning to me. In this paper, the authors compute the correlation properties (gram matrix) of the vector than went through some step of feedforward network with non linearities and random weights. Given the current interest in the theoretical description of neural nets, I think this is a paper that will be interesting for the NIPS audience and will be a welcome change between the hundreds of GAN posters. Some findings are particularly interesting.They applied their result to a memorization task and obtained an explicit characterizations of the training error. Also the fact that for some type of activation the eigenvalues of the data covariance matrix are constant suggests interesting directions for the future. Few comments and suggestions follows;: - In section 1.3 : I think that the interest for Random Network is slightly older than the authors (maybe) know. In particular, these were standard setting in the 80s (more than 20 years before any work presented by the authors) and have generated a lot of attention, especially in the statistical physics community where the generalization properties were extensivly studied, (see, just to name a few: -https://doi.org/10.1209/0295-5075/27/2/002 -https://doi.org/10.1088/0305-4470/21/1/030-10.1103/PhysRevLett.82.2975 ) - In connection with the random kitchen sinks: isn't the theory presented here a theory of what is (approximatly) going in in the Kernel space ? Given the connection between random projections and kernel methods there seems to be something interesting in this direction. Section 4.2 "Asymptotic performance of random feature methods" should propbably discuss this point. Perharps it is linked to the work of El Karoui, et al ? - Can the author provided an example of an activation that leads to the 0 value for the parameter zeta? This is not clear from the definition (10) (or maybe I missed something obvious ?) is it just ANY even function? I did not check all the computations in the supplementary materials (espcially the scary expansion in page 11!) but I am a bit confuse about the statement "t is easy to check many other examples which all support that eqn. (S19) is true, but a proof is still lacking." . Why is it called a lemma then? I am missing something?

Reviewer 2



the idea of addressing deep learning using random matrix theory seems to have a lot of potential but it appears to be in the beginning stages. Seems the major contribution is to show that their main be some advantages in keep zeta = 0 which may give future guidance when selecting and analyzing activation functions. Numerical results are provided demonstrating alignment with theory predicted. I am not sure how applicable this paper will be for deep learning in practice other than the suggested future analysis for activation functions. My background is not in random matrix theory, but the proofs appear correct as best I can tell. Minor comments: Line 73-80. Use of definitions defined in Theorem 1 before theorem 1 is presented. I would prefer a brief definition/intuition of these terms to be presented in this section. Otherwise reader needs to jump around to fully appreciate this paragraph. Line 102-103. Just English preference: "There has been minimal research performed on random matrices with nonlinear dependencies with a predominant focus on kernel random matrices." Line 122-123. Equation 4. I think it might add clarity to say that delta is the Dirac delta and make the definition depend on t: rho_M(t) = (1/n) sum delta(t-lambda_j(M)). Also, it might help to give dimensions of M in definition. Line 135-137: Equation (9) might add a reminder of how M depends on n0 -> infinity Line 214-217: Can the author point to any references/paper's why equilibrating the singular values would speed up training. I think overall it might be useful if a discussion of the practical applications for this analysis is added.

Reviewer 3



Among all the papers trying to contribute to the theory of deep learning this seems to me to be a very important contribution. This paper solves one of the open problems in random matrix theory that allows to describe spectral density of matrices that went trough a non-linearity such as used in neural nets. It is a very nice piece of work in random matrix theory with some interesting speculations about consequences for training of deep neural nets. I am very positive about interest of this paper to the NIPS crowd. Some little suggestions for improvement: Section 1.3. on related work on networks with random weights seems to be largely biased. There is a number of older works in the statistical physics community studying random neural networks (starting with works o Derrida, Gardner, Sompolinsky, some of those attempt to study multilayer networks). There is also: "Deep Neural Networks with Random Gaussian Weights: A Universal Classification Strategy?" of more recent "Multi-Layer Generalized Linear Estimation" by Manoel et al. Surely the authors can do a better job in covering the literature related to random weights. This will be interesting for the general NIPS reader. I understand that this kind of theory is still far from saying something about generalization in neural networks. But since this is such a central question perhaps the authors could comment on this so that the reader does not overlook this point. I've read the author's feedback and took it into account in my score.